# Safety and Efficacy of the Preserflo^®^ Microshunt in Refractory Glaucoma: A One-Year Study

**DOI:** 10.3390/jcm11237086

**Published:** 2022-11-29

**Authors:** Alexandre Majoulet, Benjamin Scemla, Pascale Hamard, Emmanuelle Brasnu, Alexandre Hage, Christophe Baudouin, Antoine Labbé

**Affiliations:** 1Quinze-Vingts National Ophthalmology Hospital, IHU FOReSIGHT, INSERM-DHOS CIC 1423, 75012 Paris, France; 2Department of Ophthalmology III, Quinze-Vingts National Ophthalmology Hospital, IHU FOReSIGHT, 75012 Paris, France; 3Department of Ophthalmology, Ambroise Paré Hospital, AP-HP, UVSQ, Paris Saclay University, 91190 Gif-sur-Yvette, France

**Keywords:** glaucoma, microshunt, Preserflo, refractory glaucoma, trabeculectomy

## Abstract

**Purpose:** To evaluate the safety and efficacy of Preserflo^®^ microshunt implantation in eyes with refractory glaucoma. **Methods:** In this retrospective study, a cohort of patients who underwent Preserflo^®^ microshunt implantation between April 2019 and August 2020 for refractory glaucoma were evaluated. At the time of surgery, all eyes had uncontrolled intraocular pressure (IOP) despite maximally tolerated medical therapy and at least one previous failed glaucoma filtering surgery. The primary outcome was a complete success, defined as postoperative IOP ≤ 21 mm Hg with an IOP reduction ≥ 20% and no repeat filtering surgery. The secondary outcome was qualified success, defined as a complete success with the use of antiglaucoma medications. The rates of needling, bleb repair, and postoperative complications were also recorded. **Results:** Forty-seven eyes with a mean preoperative IOP of 30.1 ± 7.1 mm Hg and a mean of 3.4 ± 1 glaucoma medications were included. The mean number of previous surgeries prior to microshunt implantation was 2.3 ± 1.3. After 1 year, the mean IOP was significantly reduced to 18.8 ± 4.6 mm Hg, with the mean number of medications significantly reduced to 1.4 ± 1.2. Complete success was achieved in 35% of eyes, and a qualified success in 60% of eyes. A decrease in IOP of at least 30% was found in 55% of eyes. Needling or bleb repair was performed in 49% of eyes. Complications were minimal and transient, except for one eye which presented with tube extrusion, and another eye with a transected tube. A repeat glaucoma surgery had to be performed in 17% of eyes. **Conclusions:** The Preserflo^®^ Microshunt provided moderate success but a significant reduction in IOP, with a good safety profile after one year of follow-up in eyes at high risk for failure of filtering surgery.

## 1. Introduction

Refractory glaucoma can be defined as uncontrolled glaucoma on maximal medical treatment after the failure of a previous filtering surgery procedure or cyclophotocoagulation [1]. Unfortunately, eyes with refractory glaucoma are at greater risk of surgical failure, mainly because of increased risk of bleb fibrosis due to excessive conjunctival scarring. In addition, a prior filtering surgery and/or previous use of MMC represents a higher risk for postoperative complications such as bleb infection or persistent hypotony [2,3,4]. The treatment options for refractory glaucoma remain repeat filtering surgery, implantation of other drainages devices, or cyclophotocoagulation. The use of drainage devices such as the Ahmed valve (New World Medical Inc., Rancho Cucamonga, CA, USA) or the Baerveldt implant (AMO Inc., Santa Ana, CA, USA) has been proven to be effective in refractory glaucoma, but these procedures are invasive and have a high rate of early and late complications (in particular, conjunctival fibrosis, choroidal detachment, or corneal endothelial decompensation) [3,5,6,7]. Laser cyclodestruction may be effective in lowering intraocular pressure (IOP), especially in multi-operated eyes, but the procedure has a low long-term success rate, and sometimes severe complications occur, such as inflammation and chronic hypotony, which can lead to phthisis [8].

In recent years, new, potentially less invasive drainage devices have been developed for the surgical management of glaucoma patients, known as Minimally Invasive Glaucoma Surgery (MIGS). MIGS are devices with a high safety profile and minimal disruption of the normal anatomy, which should have at least modest efficacy in terms of lowering IOP [9]. Although these devices are usually used as a primary glaucoma filtration surgery, their use in refractory glaucoma has been evaluated recently. In two previous studies, the Preserflo^®^ MicroShunt has shown promising 1 year safety and efficacy results in refractory glaucoma [10,11]. However, in these studies, most patients had primary open angle glaucoma, had only one filtering surgery failure, and patients with other types of prior surgeries, such as corneal transplant or retinal surgeries, were excluded.

Thus, the purpose of this study was to evaluate the 1-year safety and efficacy of the ab externo implantation of the Preserflo^®^ MicroShunt, in conjunction with MMC, on a cohort of patients with high IOP and at least one prior failed glaucoma surgery.

## 2. Materials and Methods

This study reports the results of a retrospective, single-center series of patients with refractory glaucoma who were implanted with a Preserflo^®^ MicroShunt combined with MMC 0.2 mg/mL between April 2019 and August 2020 at the Quinze-Vingts National Ophthalmology Hospital in Paris, France. This study was performed in accordance with the Declaration of Helsinki, and all patients were informed and gave their consent for data collection.

We included all patients who had undergone a Preserflo^®^ MicroShunt implantation between April 2019 and August 2020 and who had at least one prior failed glaucoma surgery (deep non-penetrating sclerectomy, trabeculectomy, Xen^®^ implant, Ahmed valve, or laser cyclophotocoagulation) on the same eye by the time of implantation. All microshunt implantations were performed by two experienced glaucoma surgeons (CB and AL). The position of the implant was chosen according to the quality of the conjunctiva, areas that had undergone previous surgeries, or the presence of synechiae of the iridocorneal angle observed on gonioscopy. Mitomycin C (MMC) at a concentration of 0.2 mg/mL was placed in the subconjunctival space for 2 to 3 min.

Patients were instructed to discontinue all glaucoma medications (topical and oral). The reintroduction of glaucoma medication, needling, bleb revision, or repeat glaucoma surgery was performed at the surgeon’s discretion.

Patient demographics, clinical and paraclinical data, including pre- and postsurgical IOP, the number of glaucoma drugs, and ophthalmological surgical history were collected. Visual field severity was classified according to the Glaucoma Staging System (GSS), and the testing was performed within the 6 months prior to surgery [12]. After the surgery, postoperative data were gathered 1 day, 1 week, 1 month, 3 months, 6 months, and 12 months later. The number of needlings and bleb revisions were also recorded. Needlings were performed at the slit lamp or in the operating room with a 25-G needle and could optionally be followed by anti-metabolite injection—5-FU (25 mg/mL) or MMC (0.1 mg/mL). If a bleb revision was required, after sub-Tenon’s local anesthesia, the conjunctiva was opened, and a careful dissection of the conjunctiva and the fibrotic Tenon’s tissue over the external portion of the microshunt was performed. MMC (0.2 mg/mL) was applied for 2 min, and, after rinsing with BSS, the conjunctiva was sutured to the limbus with absorbable sutures.

Criteria for “complete success” and “qualified success” were analyzed at 12 months. Complete success was defined as postoperative IOP ≤ 21 mm Hg, IOP reduction ≥ 20%, no repeat filtering surgery, no loss of light perception, no chronic hypotony (defined as IOP < 5 mm Hg on 2 follow-up examinations 3 months apart), and no glaucoma medications. Neither needlings nor bleb revisions were considered failures. Qualified success was defined by the same criteria as complete success, but treatment with glaucoma medication was allowed. Repeat surgeries requiring implant replacement, conversion to another filtering surgery, or loss of light perception were considered failures.

## 3. Results

Of the 58 eyes with refractory glaucoma that underwent Preserflo^®^ MicroShunt surgery, 11 were lost to follow-up. A total of 47 eyes were thus included in the study. The mean preoperative IOP was 30.11 ± 7.08 mm Hg, and the mean number of glaucoma medications was 3.40 ± 0.95. The mean visual field deviation was −19.2 ± 7.4 dB, including 74% of patients having lost more than 20 dB. The mean number of prior glaucoma surgeries was 2.3 ± 1.3. Baseline demographics are presented in Table 1.

### 3.1. Success

For an IOP range between 6 and 21 mm Hg, 35% of patients achieved complete success, and 60% of patients achieved qualified success (Figure 1). For an IOP range between 6 and 21 mm Hg, 28% of patients achieved complete success, and 36% of patients achieved qualified success. For an IOP range between 6 and 15 mm Hg, 21% of patients achieved complete success, and 28% of patients achieved qualified success. For an IOP range between 6 and 12 mm Hg, 17% of patients achieved complete success, and 10% of patients achieved qualified success.

Censored for failures (*n* = 38), the mean IOP at 1 year had decreased from 28.7 ± 7.1 mm Hg to 18.8 ± 4.6 mm Hg (−9.9 mm Hg; *p* < 10^−5^), which represents a mean decrease of 34%. 

A decrease from the initial IOP of at least 20% was found in 66% of patients, and a decrease of at least 30% in 55% of patients (Figure 2). The mean number of glaucoma medications decreased from 3.3 ± 1 to 1.4 ± 1.2 (−1.9; *p* < 10^−5^), representing a mean decrease of 64%. Fifty-one percent of patients were on no medications (Figure 3).

### 3.2. Needling or Bleb Revision

During follow-up, 49% of patients underwent at least one needling or bleb revision in the operating room. The mean number of needlings or bleb revisions was 1.7 ± 0.7. A second needling or bleb revision was required for 26% of patients. Of the patients who received a needling or bleb revision, 8% achieved complete success, 16% achieved qualified success, and 32% required further glaucoma surgery.

### 3.3. Security

A peripheral choroidal detachment (CD) was found in 8% of patients. Of these patients, only one developed persistent hypotony, requiring oral corticosteroids. All cases of CD were resolved after one month. One patient with a history of surgical retinal detachment repair presented with late macular edema. One patient presented with microshunt exposure at M2 requiring surgical revision with an amniotic membrane graft, followed by explantation at M3. No patients developed retinal detachment or malignant glaucoma (Table 2).

### 3.4. Failure

Surgery was unsuccessful in 21% of patients. A repeat glaucoma surgery had to be performed in 8 patients (17%): 5 diode lasers (11%), 1 deep sclerectomy (2%), and 2 Xen^®^ implantations (4%). One patient developed a spontaneous extrusion of his microshunt in the second month, requiring another surgery for amniotic membrane grafting and then a removal of the microshunt after 3 months. One microshunt was found transected during a bleb revision after 5 months.

## 4. Discussion

Few studies have evaluated the safety and efficacy of MIGS in refractory glaucoma. This retrospective, single-center study reports the safety and efficacy of the Preserflo MicroShunt at 1 year in patients with severe refractory glaucoma. 

Our results appear to be inferior, in terms of success, to those reported in the literature. Quaranta et al. followed 31 eyes with COAG and a previous trabeculectomy over a 12-month period [11]. In this study, complete success was defined as IOP ≤ 17 mm Hg and an IOP decrease ≥ 20% with no medical treatment. Qualified success was defined with the same criteria but with the use of one or more medications. Complete success was achieved in 67.74% of eyes, and qualified success in 95.34% of eyes. Mean IOP was significantly decreased from 24.12 ± 3.14 mm Hg to 12.56 ± 2.64 mm Hg (mean decrease of 48%). Similarly, the number of medications decreased from 3.29 ± 0.64 to 0.46 ± 0.77 (mean decrease of 82%). No eyes developed serious side effects. Durr et al. also reported the results of the Preserflo MicroShunt in 85 eyes with refractory glaucoma [10]. Complete success was achieved in 61% of patients, and qualified success was achieved in 79.7% of patients. At 12 months, the median IOP had decreased from 22.0 ± 5.5 mm Hg to 13 ± 3.5 mm Hg (41% decrease). The median number of medications decreased from 4 ± 0.5 to 0 ± 1. Severe complications were found in 6% of patients: 1 case of corneal edema, 1 case of loss of fixation, 1 case of uveal effusion, 1 retinal tear, and 1 case of endophthalmitis (not related to the Preserflo Micro-Shunt filtration bleb). There are several possible explanations for the lower success rate in our study. Quaranta et al. included only patients with open-angle glaucoma and a history of a single trabeculectomy failure. Secondary glaucomas and patients who had undergone any other type of ophthalmic surgery (aside from cataract surgery) were excluded. Durr et al. also included patients who had undergone at least one filtering surgery, of whom 24.7% of eyes had a history of at least two failed filtering surgeries. Eyes with a history of corneal or retinal surgery were excluded. In our study, the mean number of filtering surgeries was higher (2.3 ± 1.3), and 55% of patients had undergone at least two glaucoma surgeries. We also made the choice to include all types of glaucoma, as well as patients with a complicated ophthalmologic history such as corneal transplantation or retinal detachment surgery, so as to be more representative of real life. These differences could explain the poorer results observed in our study [13,14]. Indeed, factors such as predisposition to scarring, ethnicity, and inflammation leading to failure of the first surgery were still present at the time of the second surgery. In addition, conjunctival fibrosis, the presence of an already incised conjunctiva, or the use of MMC should have an even greater impact on the success of the second surgery [2,3,15]. Moreover, in the present study, the operated eyes also had more severe glaucoma. Indeed, the initial mean IOP was 30.11 ± 7.08 mm Hg compared to 22 ± 4.5 mm Hg for Durr et al. and 24.8 ± 3.86 mm Hg in the study by Quaranta et al. It has been shown that higher pre-operative IOP can be a risk factor for failure of filtering surgery [16,17,18]. Our study found a success rate of 24% in patients who received needling or bubble filtration. The reasons for the lower success rate in these patients are the same as those previously mentioned.

The position of the implant could play a role in the success of the surgery. In the study by Durr et al., 40% of Preserflo^®^ Microshunts were placed inferiorly. However, their statistical analysis found that superior or inferior location was not predictive of failure at 1 year. Another parameter that may explain the difference in our results is the duration and concentration of MMC during implantation of the Preserflo^®^ MicroShunt. In our study, MMC 0.2 mg/mL was applied for a mean of 2.13 min. Quaranta et al. chose to apply MMC for 3 min at a concentration of 0.3 mg/mL. In the study by Durr et al., the MMC was applied for 2 min, but at concentrations of only 0.2 mg/mL for 21.2% of patients and 0.4 mg/mL for 52.9% of patients, and even 0.5 mg/mL for 25.9% of patients. Durr et al. reported that a MMC dose < 0.4 mg/mL was a predictor of needling being required after microshunt surgery for refractory glaucoma [10]. In a study by the same team evaluating the efficacy of the Preserflo^®^ MicroShunt as a first-line surgery, a MMC dose of 0.2 mg/mL was a risk factor for failure (HR 2.51; 95% CI 1.12–5.65), while no patient who received a dose of 0.5 mg/mL required needling [19]. These higher MMC exposures would explain the higher success rates in previous studies. The risk–benefit ratio of increasing the MMC dose was evaluated in a meta-analysis of the Xen Gel stent^®^, a device that works similarly to the Preserflo MicroShunt^®^. It was found to increase the success rate without any effect on the needling rate. There was no increase in the number of postoperative complications [20]. However, increasing MMC concentration or time of exposure could lead to an increased risk of postoperative complications, including severe hypotony, spontaneous tube extrusion, or endophthalmitis [20]. The short follow-up period of these studies does not allow for an assessment of the long-term risk of such MMC exposure.

Interestingly, the rate of severe postoperative hypotony after implantation of a Preserflo^®^ MicroShunt appears to be lower than in other surgeries for refractory glaucoma [21]. However, several cases of spontaneous extrusion of the Preserflo^®^ MicroShunt have been reported in the literature [10,22,23]. This should lead to caution in the choice of the duration and concentration of MMC applied during surgery. One case of endophthalmitis unrelated to the Preserflo^®^ MicroShunt filtration bleb has been reported [10]. 

The efficacy of trabeculectomy after failure of one or more filtering surgeries was assessed in the study by Chen CW et al. in 59 eyes. After a follow-up period of 1 to 8 years (mean time 3 years), 14 eyes were lost to follow-up or were considered to be out of date. Of the remaining 45 eyes, 77.8% had IOP < 20 mm Hg, and 84% did not require treatment [24]. Nassiri et al. compared the safety and efficacy of a second trabeculectomy versus placement of an Ahmed valve in patients with failed trabeculectomy in 125 eyes [25]. One year of follow-up, success, defined as a decrease in IOP ≥ 20% and IOP ≤ 21 mm Hg, was achieved for 70.77% of patients in the trabeculectomy group and 70% in the Ahmed valve group. After 3 years of follow-up, the complication rate was 46.15% for the trabeculectomy group and 31.67% for the Ahmed valve group, with a significant difference in the occurrence of a filtering bleb Seidel sign (3.08% vs. 0%). These studies show a success rate similar to the results found with Preserflo and our study [10,11]. However, in the TVT study, a total of 212 eyes of 212 patients were enrolled, including 107 in the tube group and 105 in the trabeculectomy group. At one year, IOP (mean ± SD) was 12.4 ± 3.9 mm Hg in the tube group and 12.7 ± 5.8 mm Hg in the trabeculectomy group (*p* = 0.73). The number of glaucoma medications (mean ± SD) was 1.3 ± 1.3 in the tube group and 0.5 ± 0.9 in the trabeculectomy group (*p* < 0.001). The cumulative probability of failure during the first year of follow-up was 3.9% in the tube group and 13.5% in the trabeculectomy group (*p* = 0.017). This could be explained by the higher baseline IOP in our study (30.11 ± 7.08 vs. 25.1 ± 5.3 in tube group and 25.6 ± 5.3 in trabeculectomy group) and more previous intraocular surgery (2.3 1.3 glaucoma surgery vs. 1.3 ± 0.5 trabeculectomy and/or cataract surgery in the tube group and 1.2 ± 0.5 in the trabeculectomy group) [26]. On the other hand, the complication rate appears to be higher with trabeculectomy or Ahmed valve than with Preserflo, which would suggest a higher safety rate following Preserflo implantation. However, the long-term safety profile of Preserflo is still unclear and remains to be determined.

A recent review of the literature on cyclophotocoagulation reported an efficacy rate, defined as IOP between 5 and 21 mm Hg, ranging from 50 to 83.7% for a range of 1.16 to 1.86 treatments per eye. The follow-up time of these studies ranged from 6 to 80 months. The main complications found were hypotony (mean of 10% of cases), chronic inflammation (mean 10% of cases), and a decrease in visual acuity secondary to cataract (4.8% to 10% of cases) or macular edema (1 to 12.5%) [8]. As for the micropulsed diode, it showed a decrease in IOP of about 40% for a follow-up period of 6 to 12 months. The complication rate was very low. Mydriasis was found in 1 to 15% of cases. For these two techniques, a significant decrease in IOP was noted, but there was a high rate of reoperation, reaching as high as 80% depending on the population and the surgical protocols.

A multicenter, intention-to-treat study in 65 eyes investigated the efficacy of Xen in refractory glaucoma, defined as failure of a glaucoma procedure (laser or surgery) or uncontrolled IOP on maximal therapy (85.6% of eyes had a history of glaucoma surgery or diode laser) [27]. At 12 months, IOP had decreased from 25.1 ± 3.7 mm Hg to 15.9 ± 5.2 mm Hg (mean decrease of 37%), and 75% of patients had an IOP decrease ≥ 20%. Fewer glaucoma medications were required for 69.2% of patients, while 30.8% required a similar number of medications. Nine patients (14%) required a repeat glaucoma procedure. No serious side effects were reported.

Another study similarly evaluated the efficacy of Xen after a failed first glaucoma surgery in 72 eyes, with a median follow-up time of 26 months (6–50 months) [28]. Mean IOP had decreased from 24.82 ± 8.03 mm Hg to 17.45 ± 5.84 mm Hg (mean decrease of 23%). At 24 months, the complete success rate was 13.4% (IOP ≤ 15 mm Hg, 20% decrease in IOP without treatment), and the qualified success rate was 59.4% (IOP ≤ 15 mm Hg, 20% decrease in IOP, and less than two glaucoma medications). Eleven eyes (15.2%) underwent repeat glaucoma surgery, and 33% of patients had postoperative complications. These results show a similar success rate of Xen compared to Preserflo but with a heterogeneous safety profile depending on the study.

Limitations of our study include the retrospective, single-center design, the small sample size, and the short duration of follow-up.

## 5. Conclusions

In conclusion, Preserflo implantation in a severe, refractory glaucoma population shows a moderate success rate, but a significant decrease in IOP and number of glaucoma medications, with a good safety profile. The Preserflo MicroShunt could be a good choice for vulnerable or multioperated eyes at high risk for complications. However, further prospective studies and comparisons with other surgical techniques for refractory glaucoma are needed to support our results.

## Figures and Tables

**Figure 1 jcm-11-07086-f001:**
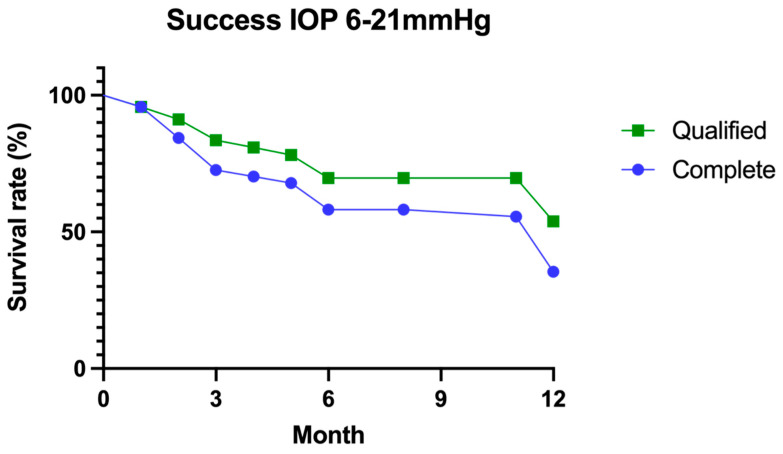
Qualified and complete success at 12 months.

**Figure 2 jcm-11-07086-f002:**
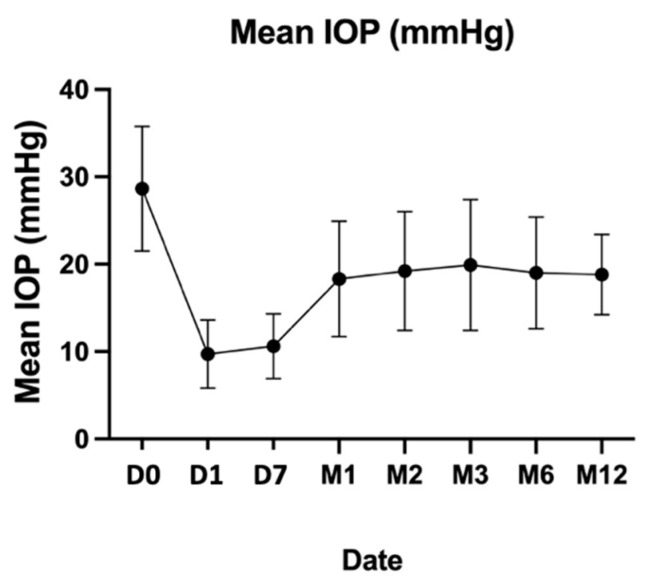
Mean IOP at 12 months.

**Figure 3 jcm-11-07086-f003:**
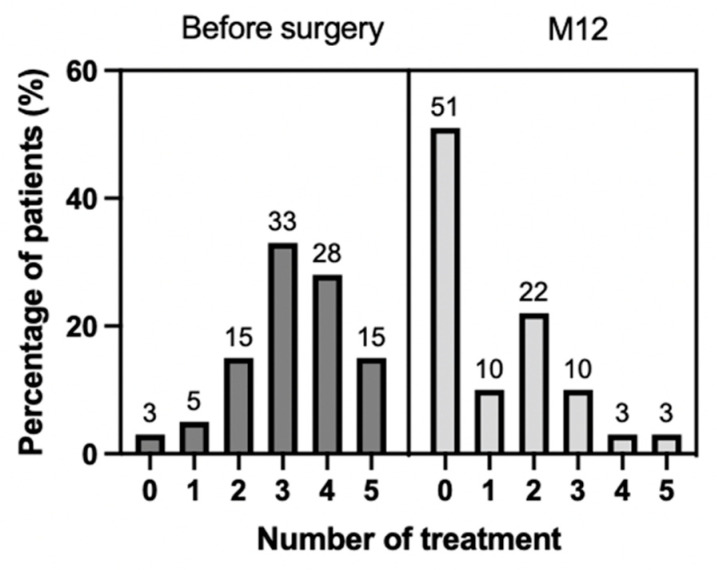
Number of treatment at 12 months.

**Table 1 jcm-11-07086-t001:** Population.

Population	Numbers
Mean age (years)	64
Women	38%
Mean BCVA (logMar)	0.84 ± 0.7
Mean pre-op IOP (mm Hg)	30.11 ± 7.08
Mean pre-op number of medications	3.40 ± 0.95
Glaucoma etiology	
Primary	25 (54%)
Pseudoexfoliation	4 (9%)
Pigmentary	4 (9%)
Juvenile	1 (2%)
Congenital	3 (6%)
Uveitic	3 (6%)
Silicone oil	2 (4%)
Other	5 (10%)
Glaucoma severity	
Mild	5 (11%)
Moderate	7 (15%)
Advanced	35 (74%)
Cup/disc (mean)	0.9
Mean pre-op. MD (dB)	−19.2 ± 7.4
Surgical history	
Mean number of glaucoma surgeries	2.3 ± 1.3
Trabeculectomy or Deep Sclerectomy (nb)	45 (96%)
Xen^®^	8 (17%)
Ahmed	2 (4%)
Starflo^®^	3 (6%)
Cypass^®^	1 (2%)
Diode	13 (28%)
Cataract	42 (89%)
Retinal-Detachment—vitrectomy	4 (8%)
Corneal transplant	2 (4%)
Other	2 (4%)
Mean intraoperative MMC (min)	2.13

BCVA: Best Corrected Visual Acuity, IOP: Intraocular Pressure, MD: Mean Deviation of the visual field, MMC: mitomycin C.

**Table 2 jcm-11-07086-t002:** Early and late complications.

Complication	Early (<3 Months)	Late (≥3 Months)
Choroidal detachment	4	
Dellen effect	1	
Tube exposure	1	
Tube transection		1
Macular edema		1

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
