# Peer review of "Safety and Efficacy of the Preserflo® Microshunt in Refractory Glaucoma: A One-Year Study"

_jcm, 2022, doi:10.3390/jcm11237086_

Round 1
Reviewer 1 Report
To Dear Author
Please find my comments:
1- There are some minor corrections required for English of manuscript.
2- Material and method section is too long. Please brief it as possible.
3- Please correct the mentioned parameter (Women) applied in row in table 3.
4- I suggest to merge a and b charts in figure 1. you can use different font size or color to differ "Qualified Success" and "Complete Success".
5- It seems that the failure rate of surgery is high. What is the cause of it? Also, table 3 is not informative. You can include its data in manuscript.
6- Due to the concept of your experiment, I suggest to add "limitation of study".
Reviewer 2 Report
The present study evaluated the safety and efficacy of Preserflo® microshunt implantation in eyes after at least one failed prior glaucoma surgery (deep non-penetrating sclerectomy, trabeculectomy, Xen® implant, Ahmed valve, 80 or laser cyclophotocoagulation). According to results of the study complete success was achieved in 35% of eyes, and a qualified success in 60% of eyes.
As a general comment Preserflo® microshunt is a promising new bleb forming device and the study adds some new data to the subject.
I have several comments:
1. More than half of the patients (54%) had advanced glaucoma were lower IOPs are necessary. Based on the glaucoma advanced status, success (qualified and complete) should additionally calculated for different IOP upper limits, eg as following: ≤ 18 mmHg, ≤ 15 mmHg, ≤ 12 mmHg.
2. Purpose the purpose of this study was to evaluate the 1-year safety and efficacy……….. with severe refractory glaucoma with high IOP and a mean of 2 prior failed glaucoma surgeries. Methods We included all patients who had undergone a Preserflo® MicroShunt implantation be tween April 2019 and August 2020 and who had at least one failed prior glaucoma surgery. This should be corrected
3. Methods and postoperative IOP at D1, D7, D30, M2, M3, M6, and M12, optic nerve cup/disc ratio, 102 number and frequency of glaucoma medications, visual acuity, severity of visual field 103 (VF) damage, mean retinal sensitivity deviation assessed by VF, short-term complications and number of needlings or bleb revisions were also recorded. The authors should explain abbreviations and should indicate in which visit each exam was performed
4. Most of the patients included had trabeculectomy (96%) and cataract (89%) as primary surgeries. Therefore, in the discussion section the authors may compare their results with TVT study.
5. The authors report 24% success of their operations requiring needling. This needs to be discussed.
6. The efficacy of trabeculectomy after failure of one or more filtering surgeries was 246 assessed in the study of Chen CW et al. in 59 eyes. After a follow-up period of 1 to 8 years 247 (mean time 3 years), 14 eyes were lost to follow-up or were considered to be out of date. 248 Of the remaining 45 eyes, 77.8% had IOP < 20 mmHg, and 84% did not require treatment 249 (24).24 does not apply
